# ADAPTIVE PARAMETER-EFFICIENT FINE-TUNING VIA MULTI-TASK OPTIMIZATION ON SUBSET SELECTION

## ABSTRACT

Parameter-efficient fine-tuning (PEFT) is a highly effective approach for adapting large pre-trained models to downstream tasks with minimal computational overhead. At the core, PEFT methods freeze most parameters and only trains a small subset (say $< 0.1\%$ of total parameters). Notably, different PEFT methods select different subsets of parameters and result in varying performances. This variation prompts a key question: how to adaptively select the most influential subset?

We formulate the subset selection as a multi-task problem: maximizing the performance and minimizing the number of trainable parameters, which consists of both discrete and continuous objectives. We leverage a series of transformations – including $\epsilon$-constraint method and second-order Taylor approximation – to arrive at the classical 0-1 knapsack problem, which we solve via the lens of Pareto optimality. Consequently, we propose AdaPEFT, an efficient and scalable algorithm for PEFT that adapts to various tasks, in which our subset selection is consistent as the training horizons and model sizes scale up over $50\times$.

## 1 INTRODUCTION

Fine-tuning is an important technique in deep learning, which adapts large pre-trained models to new tasks quickly. At high level, there are two classes of fine-tuning methods: (I) fine-tune the entire model (i.e. 100% of parameters are trainable), or (II) only update a small portion of the model (e.g. 0.1%) and freeze the majority of parameters. The second class of methods is known as parameter-efficient fine-tuning (PEFT), including examples such as LoRA Hu et al. (2022), prompt tuning Lester et al. (2021), linear probing (or last-layer tuning), LayerNorm tuning Zhao et al., and BitFit Zaken et al. (2022). In contrast to full-model fine-tuning (FMT), PEFT enjoys $\approx 50\%$ speedup and significantly reduced memory cost, e.g. LoRA uses $5\times$ less memory than FMT on LLAMA2-7B Li et al. (2024).

On the other hand, the performance of PEFT depends on the choice of methods, or the choice of a subset of parameters. With a proper choice of trainable parameters, PEFT can be as performant as FMT. For example, training RoBERTa-base on 8 datasets in the GLUE benchmark Wang et al. (a), FMT gets an average accuracy 86.4%, LoRA gets 87.2%, and BitFit gets 85.2%(see Table 2 in Hu et al. (2022)); training GPT3 on WikiSQL Zhong et al. (2017), FMT gets an accuracy 73.8%, LoRA gets 73.4%, and BitFit gets 71.3% (see Table 4 in Hu et al. (2022)). However, the success of these PEFT methods may not be reproduced in some tasks. For example, LoRA significantly underperforms FMT on CoLA and MRPC datasets (see Table 1 of Wang et al. (2024b; b)); on coding and mathematics domains, Biderman et al. (2024) shows that "LoRA substantially underperforms full finetuning" in both instruction tuning and continued learning ($\approx 20B$ tokens).

As a consequence, we study the following question:

> *Q: How to adaptively select parameter groups (or active subset of parameters) so that the model achieves high utility while only has a small portion of trainable parameters?*

To answer this question, we build on top of existing PEFT methods and formulate a **multi-task problem**. More precisely, we minimize the loss as a bi-level objective where the upper-level objective

is **discrete** and the low-level one is **continuous**, and we minimize the number of trainable parameters as a purely **discrete** objective over the subsets:

$$\min_{\mathcal{A}} \left( L_{\mathcal{A}} := \min_{\boldsymbol{w}_{(k)} \in \mathcal{A}} L(\boldsymbol{w}_{(1)}, ..., \boldsymbol{w}_{(K)}) \right), \quad \min_{\mathcal{A}} \left( \frac{|\mathcal{A}|}{|\boldsymbol{w}|} := \frac{\sum_k \mathbb{I}(\boldsymbol{w}_{(k)} \in \mathcal{A}) \cdot |\boldsymbol{w}_{(k)}|}{\sum_k |\boldsymbol{w}_{(k)}|} \right) \quad (1)$$

Here $\boldsymbol{w} := \{\boldsymbol{w}_{(1)}, \boldsymbol{w}_{(2)}, ..., \boldsymbol{w}_{(K)}\}$ is the set of all parameter groups in a model (inspired by existing PEFT methods), i.e. $|\boldsymbol{w}| = \sum_k |\boldsymbol{w}_{(k)}|$; $L$ is the loss function; $\mathcal{A}$ is the active set of parameter groups with $|\mathcal{A}| = \sum_k \mathbb{I}(\boldsymbol{w}_{(k)} \in \mathcal{A})|\boldsymbol{w}_{(k)}|$, e.g. $\mathcal{A} := \{\boldsymbol{w}_{(1)}, \boldsymbol{w}_{(3)}\}$. As shown in Table 1, each PEFT method corresponds to some fixed subset $\mathcal{A} \subseteq \boldsymbol{w}$, and we recover the FMT when $\mathcal{A} = \boldsymbol{w}$.

Table 1: Summary of PEFT methods (row) and corresponding parameter groups (column). Here 'lora_A/lora_B' are low-rank matrices, 'head' is the last linear layer, 'norm' is layer normalization, and 'bias' is bias terms. Y/N indicate whether a parameter group is active and trainable.

| | lora_A | lora_B | head | norm | bias |
|---|---|---|---|---|---|
| LoRA | Y | Y | N | N | N |
| LoRA-FA | N | Y | N | N | N |
| BitFit | N | N | N | N | Y |
| Linear probing | N | N | Y | N | N |
| LayerNorm | N | N | N | Y | N |
| AdaPEFT (ours) | adaptive | | | | |

In contrast to applying a fixed $\mathcal{A}$ like existing PEFT, we enable the **adaptivity** of $\mathcal{A}$ in (1). However, the two tasks in (1) may conflict with each other. For instance, the minimum number of trainable parameters is 0 but then the model is not trainable at all, hence the loss is not minimized.

To resolve the conflict, we solve our multi-task problem in (1) under the **Pareto optimality**.

**Definition 1.1.** (Pareto optimality for PEFT). For two active sets $\mathcal{A}_1$ and $\mathcal{A}_2$, if $L_{\mathcal{A}_1} \leq L_{\mathcal{A}_2}$ and $|\mathcal{A}_1| \leq |\mathcal{A}_2|$ with at least one inequality being strict, then $\mathcal{A}_1$ dominates $\mathcal{A}_2$. Furthermore, an active set is Pareto optimal if it is not dominated by any other sets.

**Contribution.**

- We formulate a multi-task problem with both discrete and continuous objectives, to adaptively select the active set of parameter groups for PEFT under the Pareto optimality.

- We transform our multi-task problem (optimized on both subsets and parameters) to a classical single-task problem (optimized on binary variables) that is known as 0-1 knapsack problem. Specifically, our transformation leverages a series of methods including $\epsilon$-constraint method, gradient descent, and Taylor approximation.

- We propose efficient algorithms to compute the Hessian-informed loss reduction (without extra back-propagation) and to solve the 0-1 knapsack problem.

- We observe consistent patterns indicating that influential parameter groups can be discovered early in training, and such patterns can transfer across model sizes. These scaling patterns serve as the basis for our adaptive PEFT (AdaPEFT; see Figure 1).

**Related work.** We briefly discuss some related work and extensively discuss them in Appendix E. Broadly speaking, our multi-task framework can be formulated on all subset-based PEFT methods and their combinations, because it is adaptive to model architectures and tasks. However, most multi-task methods such as GradNorm Chen et al. (2018), MGDA Désidéri (2012); Sener & Koltun (2018), and PCGrad Yu et al. (2020) cannot be applied directly since the subset optimization (1) is discrete and the meta-minimization is bi-level. As a solution, we work on the 0-1 knapsack problem by drawing on techniques from Bu & Xu (2024) to compute Hessian-informed loss reduction, and employing a greedy approximation algorithm Martello & Toth (1990) to estimate the Pareto frontier.

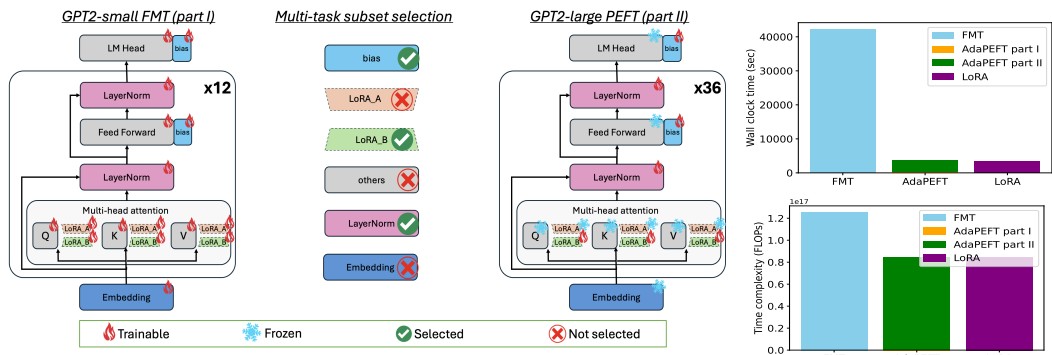

Figure 1: AdaPEFT framework. Left: Illustrated of Algorithm 2 on GPT2. Right: wall-clock time (upper) and time complexity (lower) for fine-tuning GPT2-large on three methods (see details in Appendix D). For AdapPEFT, part I refers to FMT with Algorithm 1 on GPT2-small for 10% of the total traning horizon and part II refers to running the PEFT with our selected subsets.

## 2 PROBLEM FORMULATION

Table 2: Roadmap of transformations from multi-task optimization to 0-1 knapsack problem.

| reference | multi-task | problems | constrained | bi-level | transformation |
|---|---|---|---|---|---|
| (1) | yes | minimization on (subset, $\boldsymbol{w}$) | no | yes | $--$ |
| (2) | no | minimization on (subset, $\boldsymbol{w}$) | yes | yes | $\epsilon$-constraint |
| (7) | no | minimization on (subset, $\eta$) | yes | yes | gradient descent |
| (10) | no | maximization on subset | yes | no | Taylor approximation |
| (11) | no | maximization on binary | yes | no | knapsack problem |

### 2.1 NOTATIONS

We denote $\boldsymbol{w} \in \mathbb{R}^D$ as all model parameters, while $\boldsymbol{w}_t$ represents the iteration $t$ and $\boldsymbol{w}_{(k)}$ represents the $k$-th parameter group. The same notation follows for other variables including the mini-batch gradient $\mathbf{g} \in \mathbb{R}^D$. We denote the loss as $L(\boldsymbol{w})$, its first-order derivative as $\mathbf{G}_{(k)} := \frac{\partial L}{\partial \boldsymbol{w}_{(k)}}$ and its second-order derivative as $\mathbf{H}_{(k)} := \frac{\partial^2 L}{\partial \boldsymbol{w}_{(k)}^2}$. We omit $t$ when it is obvious from the context.

### 2.2 MULTI-TASK OPTIMIZATION AND PARETO OPTIMALITY

In this section, we transform the multi-task problem to a single-task problem, which has a bi-level objective and a constraint. We use the $\epsilon$-constraint method as a scalarization technique, which chooses one task to optimize and converts the remaining tasks into constraints. Thus, (1) leads to

$$\min_{\mathcal{A}} L_{\mathcal{A}} \quad \text{s.t.} \quad |\mathcal{A}|/|\boldsymbol{w}| \le \epsilon \tag{2}$$

However, we emphasize that a solution to (2) is not necessarily Pareto optimal in terms of (1) (shown in Theorem 2.1), unless we restrict ourselves to an unrealistic case where (2) has a unique solution.

**Theorem 2.1.** *For any $\epsilon \ge 0$, a solution to* (2) *may not be Pareto optimal of* (1). *Nevertheless, if* (2) *has only one solution, then the solution is Pareto optimal.*

*Proof.* Suppose $\mathcal{A}^\star$ is a solution to (2) but not Pareto optimal, then $\exists \mathcal{A}'$ such that,

$$L_{\mathcal{A}'} \le L_{\mathcal{A}^\star} \text{ and } |\mathcal{A}'| \le |\mathcal{A}^\star| \tag{3}$$

with at least one strict inequality. Hence $\mathcal{A}'$ is also in the feasible domain since $|\mathcal{A}'| \le |\mathcal{A}^\star| \le \epsilon|\boldsymbol{w}|$. Now $\mathcal{A}^\star$ being the minimizer of (2) gives $L_{\mathcal{A}^\star} \le L_{\mathcal{A}'}$. Hence, $L_{\mathcal{A}'} = L_{\mathcal{A}^\star}$, and the strict inequality can only be $|\mathcal{A}'| < |\mathcal{A}^\star|$. If the solution of (2) is unique, contradiction; otherwise, $\mathcal{A}^\star$ may not be Pareto optimal. $\square$

To guarantee the Pareto optimality when (2) has more than one solutions, we propose a **refined solution** in two steps: (I) exhaustively find all solutions of (2), denoted by the set $\operatorname{argmin}_{\mathcal{A}:|\mathcal{A}|/|\boldsymbol{w}|\leq\epsilon}L_{\mathcal{A}}$; (II) select one solution with the smallest number of trainable parameters. Mathematically, we select

$$\mathcal{A}^{\star}(\epsilon) = \operatorname{argmin}_A \left\{ |A| : A \in \operatorname{argmin}_{\mathcal{A}:|\mathcal{A}|/|\boldsymbol{w}|\leq\epsilon}L_{\mathcal{A}} \right\} \tag{4}$$

**Theorem 2.2.** *For any $\epsilon \geq 0$, the refined solution to (2) (i.e. (4)) is always Pareto optimal of (1).*

*Proof.* Similar to the proof of Theorem 2.1, if $\exists \mathcal{A}'$ that dominates $\mathcal{A}^{\star}$, then $L_{\mathcal{A}'} = L_{\mathcal{A}^{\star}}$ and $|\mathcal{A}'| < |\mathcal{A}^{\star}|$. Hence $\mathcal{A}'$ belongs to the solution set, which contradicts that $\mathcal{A}^{\star}$ has the smallest $|\mathcal{A}|$ among all solutions. $\square$

*Remark* 2.3. Theorem 2.1 and Theorem 2.2 hold without requiring convexity. Note each $\epsilon$ yields one $\mathcal{A}^{\star}(\epsilon)$, hence one can sweep through $0 \leq \epsilon \leq 1$ to get different Pareto optimal solutions.

## 2.3 GRADIENT DESCENT FOR LOW-LEVEL MINIMIZATION

In this section, we transform the bi-level meta-minimization to a single-level problem, by providing a closed-form approximation for the low-level minimization of (2), i.e. $L_{\mathcal{A}} := \min_{\boldsymbol{w}_{(k)}\in\mathcal{A}} L(\boldsymbol{w})$. This is achieved by two steps: (I) translating the minimization over $\boldsymbol{w} \in \mathbb{R}^D$ to the minimization over $\eta \in \mathbb{R}$ and (II) using the Hessian-informed solution to explicitly solve the minimization.

Algorithmically, the low-level minimization is solved iteratively via gradient descent, such as SGD and Adam. For example, for $1 \leq t \leq T$,

$$\boldsymbol{w}_{(k),t+1} = \boldsymbol{w}_{(k),t} - \eta I_{(k)}\mathbf{g}_{(k),t} := \boldsymbol{w}_{(k),t} - \eta\mathbb{I}(\boldsymbol{w}_{(k)} \in \mathcal{A})\mathbf{g}_{(k),t} \tag{5}$$

in which the binary mask $\mathbb{I}(\boldsymbol{w}_{(k)} \in \mathcal{A})$ assigns 0 to frozen groups and 1 to active groups. Note (5) recovers FMT: $\boldsymbol{w}_{t+1} = \boldsymbol{w}_t - \eta\mathbf{g}_t$ since $\mathbb{I}(\boldsymbol{w}_{(k)} \in \boldsymbol{w}) \equiv 1$.

As a consequence, the gradient descent translates the high-dimensional parameter minimization $\min_{\boldsymbol{w}_{(k)}\in\mathcal{A}} L(\boldsymbol{w})$ to an one-dimensional hyperparameter $\min_{\eta\in\mathbb{R}} L(\boldsymbol{w}_T)$:

$$L_{\mathcal{A}} \approx \min_{\eta} L(\boldsymbol{w}_T) \text{ s.t. (5)} \tag{6}$$

Furthermore, the optimization problem $\min_{\boldsymbol{w}} L(\boldsymbol{w})$ is unconstrained, whereas the optimization $\min_{\eta \text{ s.t. (5)}} L(\boldsymbol{w}_T(\eta))$ is restricted to the gradient descent path $\{\boldsymbol{w}_t\}_t$ governed by the update rule (5). We know $\min_{\eta \text{ s.t. (5)}} L(\boldsymbol{w}_T(\eta)) \geq \min_{\boldsymbol{w}} L(\boldsymbol{w})$, with the equality holding if and only if a global minimizer of $L$ lies on the path $\{\boldsymbol{w}_t\}_t$.

To enjoy a closed-form approximation, we study the minimization from a local perspective (i.e. one iteration at a time) where (6) is equivalent to

$$L_{\mathcal{A}} \approx \sum_{t=1}^{T-1} \min_{\eta}[L(\boldsymbol{w}_{t+1};\eta) - L(\boldsymbol{w}_t)] \text{ s.t. (5)} \tag{7}$$

up to a constant $L(\boldsymbol{w}_0)$. At each iteration, by only updating one parameter group (say $\mathcal{A} = \{\boldsymbol{w}_{(k)}\}$) and freezing all the others, the loss reduction from this $\boldsymbol{w}_{(k)}$ is

$$L(\boldsymbol{w}_{t+1};\eta) - L(\boldsymbol{w}_t) \approx \Delta L_{(k),t}(\eta) := -\eta\mathbf{G}_{(k),t}^{\top}\mathbf{g}_{(k),t} + \frac{\eta^2}{2}\mathbf{g}_{(k),t}^{\top}\mathbf{H}_{(k),t}\mathbf{g}_{(k),t} \tag{8}$$

We note that the second-order Taylor approximation is reasonably accurate, since the approximation error is $o(\eta^2)$ and hence negligible in practice (c.f. Figure 2 in Bu & Xu (2024)).

Therefore, if $\mathbf{g}_{(k)}^{\top}\mathbf{H}_{(k)}\mathbf{g}_{(k)} > 0$, updating the $k$-th parameter group can minimize (8) to $-\frac{(\mathbf{G}_{(k)}^{\top}\mathbf{g}_{(k)})^2}{\mathbf{g}_{(k)}^{\top}\mathbf{H}_{(k)}\mathbf{g}_{(k)}}$ under $\eta = \frac{\mathbf{G}_{(k)}^{\top}\mathbf{g}_{(k)}}{\mathbf{g}_{(k)}^{\top}\mathbf{H}_{(k)}\mathbf{g}_{(k)}}$. Extending to multiple parameter groups, we write the total loss reduction as

$$L_{\mathcal{A}} \approx \sum_{k,t} \Delta L_{(k),t} = -\sum_{k}\mathbb{I}(\boldsymbol{w}_{(k)} \in \mathcal{A}) \cdot \sum_{t} \frac{\left(\mathbf{G}_{(k),t}^{\top}\mathbf{g}_{(k),t}\right)^2}{\mathbf{g}_{(k),t}^{\top}\mathbf{H}_{(k),t}\mathbf{g}_{(k),t}} \tag{9}$$

which explicitly replaces the low-level minimization problem of (7).

## 2.4 KNAPSACK PROBLEM

All in all, we transform the meta-minimization in (2) to a subset maximization problem in (10),

$$\min_{\mathcal{A}} \left( \min_{\boldsymbol{w}_{(k)} \in \mathcal{A}} L(\boldsymbol{w}) \right) \approx \max_{\mathcal{A}} \sum_k \mathbb{I}(\boldsymbol{w}_{(k)} \in \mathcal{A}) \cdot \sum_t \frac{\left( \mathbf{G}_{(k),t}^\top \mathbf{g}_{(k),t} \right)^2}{\mathbf{g}_{(k),t}^\top \mathbf{H}_{(k),t} \mathbf{g}_{(k),t}} \text{ s.t. } |\mathcal{A}|/|\boldsymbol{w}| \le \epsilon \quad (10)$$

Finally, given that $\mathcal{A}$ is only reflected in the binary variable $\mathbb{I}(\boldsymbol{w}_{(k)} \in \mathcal{A})$, we can further simplify the subset maximization in (10) to a binary maximization problem:

$$\max_{I_k \in \{0,1\}} \sum_k I_k \cdot \sum_t \frac{\left( \mathbf{G}_{(k),t}^\top \mathbf{g}_{(k),t} \right)^2}{\mathbf{g}_{(k),t}^\top \mathbf{H}_{(k),t} \mathbf{g}_{(k),t}} \text{ s.t. } \frac{\sum_k I_k |\boldsymbol{w}_{(k)}|}{\sum_k |\boldsymbol{w}_{(k)}|} \le \epsilon \quad (11)$$

Importantly, (11) is essentially the 0-1 knapsack problem in Definition 2.4, which is an NP-complete combinatorial problem.

**Definition 2.4.** (0-1 knapsack problem). Given a set of items, each with a weight $W_k$ and a value $V_k$, the knapsack problem determines which items to include (whether $I_k = 1$ so that the total value is maximized within the total weight limit $W_{\text{limit}}$:

$$\max_{I_k \in \{0,1\}} \sum_k V_k \times I_k \text{ s.t. } \sum_k W_k \times I_k \le W_{\text{limit}} \quad (12)$$

From the perspective of PEFT, we view each parameter group as an item, the parameter count as weight $W_k := |\boldsymbol{w}_{(k)}|$ and loss reduction as value $V_k := \sum_t \frac{\left( \mathbf{G}_{(k),t}^\top \mathbf{g}_{(k),t} \right)^2}{\mathbf{g}_{(k),t}^\top \mathbf{H}_{(k),t} \mathbf{g}_{(k),t}}$, under the limit $W_{\text{limit}} := \epsilon \sum_k |\boldsymbol{w}_{(k)}|$.

With hindsight, (11) is equivalent to applying $\epsilon$-constraint method to the following multi-task optimization,

$$\max_{\mathcal{A}} \sum_k \mathbb{I}(\boldsymbol{w}_{(k)} \in \mathcal{A}) \cdot \sum_t \frac{\left( \mathbf{G}_{(k),t}^\top \mathbf{g}_{(k),t} \right)^2}{\mathbf{g}_{(k),t}^\top \mathbf{H}_{(k),t} \mathbf{g}_{(k),t}}, \quad \min_{\mathcal{A}} \frac{\sum_k \mathbb{I}(\boldsymbol{w}_{(k)} \in \mathcal{A}) \cdot |\boldsymbol{w}_{(k)}|}{\sum_k |\boldsymbol{w}_{(k)}|}. \quad (13)$$

## 3 ALGORITHMS

We discuss two classes of algorithms – one to efficiently compute the objectives of the knapsack problem, and the other to actually solve the knapsack problem.

### 3.1 EFFICIENTLY COMPUTING LOSS REDUCTION WITHOUT BACK-PROPAGATION

We propose Algorithm 1 to efficiently compute the loss reduction, which requires the knowledge of $\mathbf{G}_{(k)}^\top \mathbf{g}_{(k)}$ and $\mathbf{g}_{(k)}^\top \mathbf{H}_{(k)} \mathbf{g}_{(k)}$. We use two techniques, (I) quadratic curve fitting and (II) lazy updating.

**Quadratic curve fitting.** We fit the quadratic function (8) by using multiple forward passes at different learning rates to get different $\Delta L_{(k)}(\eta)$. Next, we directly find the two scalars $\mathbf{G}_{(k)}^\top \mathbf{g}_{(k)}$ and $\mathbf{g}_{(k)}^\top \mathbf{H}_{(k)} \mathbf{g}_{(k)}$ via a finite-sum problem:

$$\mathbf{g}_{(k)}^\top \mathbf{H}_{(k)} \mathbf{g}_{(k)}, \mathbf{G}_{(k)}^\top \mathbf{g}_{(k)} \approx \text{argmin}_{A,b} \sum_j \left( \Delta L_{(k)}(\hat{\eta}_j) + \hat{\eta}_j b - \frac{\hat{\eta}_j^2}{2} A \right)^2$$

where $\hat{\eta}_j \in \{-2\eta, -\eta, 0, \eta, 2\eta\}$. This back-propagation-free approach from Bu & Xu (2024) has minimal memory overhead, as it does not instantiate $\mathbf{G}_{(k)}$ or $\mathbf{H}_{(k)}$.

**Lazy updating.** We need $4K + 1$ forward passes, hence incurring $O(K)$ training time overhead if implemented naively. This overhead can be reduced significantly if the loss reduction is computed infrequently, following Bu & Xu (2024). In practice, we update every $O(K)$ iterations so that the overhead is $O(1)$ and roughly negligible.

---

**Algorithm 1** Hessian-informed loss reduction (iteration $t$)

---

1: Compute loss $L_0 = L(\boldsymbol{w}_t)$ by forward pass
2: Compute gradient $\mathbf{g} := \{\mathbf{g}_{(k)}\}_k$ on $L_0$ by SGD, AdamW, etc.
3: **if** $t \bmod 4K == 0$ **then**
4:     **for** $k \in 1, ..., K$ **do**
5:         **for** $\hat{\eta}_j \in \{-2, -1, 1, 2\} \cdot \eta_t$ **do**
6:             Compute $L_j = L\left(\boldsymbol{w}_t - \hat{\eta}_j \boldsymbol{e}_k \odot \mathbf{g}\right)$ by forward pass
7:         Fit a quadratic curve from $\{\hat{\eta}_j\} \rightarrow \{L_j - L_0\}$
8:         Derive $\mathbf{G}_{(k),t}^\top \mathbf{g}_{(k),t}$ and $\mathbf{g}_{(k),t}^\top \mathbf{H}_{(k),t} \mathbf{g}_{(k),t}$ in (8)
9:         **if** $\mathbf{G}_{(k),t}^\top \mathbf{g}_{(k),t} > 0, \mathbf{g}_{(k),t}^\top \mathbf{H}_{(k),t} \mathbf{g}_{(k),t} > 0, \text{R2 score} > 0.99$ **then**
10:             Accumulate $\text{APPI}_k = \text{APPI}_k + \frac{|\mathbf{G}_{(k),t}^\top \mathbf{g}_{(k),t}|^2}{\mathbf{g}_{(k),t}^\top \mathbf{H}_{(k),t} \mathbf{g}_{(k),t} \cdot |\boldsymbol{w}_{(k)}|}$
11:             Derive optimal learning rates $\eta_{(k)}^* = \frac{\mathbf{G}_{(k),t}^\top \mathbf{g}_{(k),t}}{\mathbf{g}_{(k),t}^\top \mathbf{H}_{(k),t} \mathbf{g}_{(k),t}}$
12: Update the parameters: $\boldsymbol{w}_{(k),t+1} = \boldsymbol{w}_{(k),t} - \eta_{(k)}^* \mathbf{g}_{(k),t}$

---

### 3.2 SOLVING 0-1 KNAPSACK PROBLEM

**Exhaustive search** To obtain the Pareto optimal solution of (13), we need to exhaustively search all $2^K$ subsets and compute the value $\sum_k \mathbb{I}(\boldsymbol{w}_{(k)} \in \mathcal{A}) \cdot V_k$ and the weight $\sum_k \mathbb{I}(\boldsymbol{w}_{(k)} \in \mathcal{A}) \cdot W_k$ for each subset. Then we find all subsets with $|\mathcal{A}|/|\boldsymbol{w}| \leq \epsilon$ and select the subset with the largest value. This algorithm guarantees to find a Pareto optimal minimizer by Theorem 2.2, but it has an exponential complexity in terms of $K$.

**Approximate solutions** In practice, we turn to approximate solutions like the greedy approximation Martello & Toth (1990) to solve (11) instead of directly (13). We define the value-to-weight ratio as *Per-Parameter Influence* (PPI) and its accumulation where

$$\text{PPI}_k(t) = \frac{(\mathbf{G}_{(k),t}^\top \mathbf{g}_{(k),t})^2}{\mathbf{g}_{(k),t}^\top \mathbf{H}_{(k),t} \mathbf{g}_{(k),t} \cdot |\boldsymbol{w}_{(k)}|}, \quad \text{APPI}_k(\tau) = \sum_{t=1}^{\tau} \text{PPI}_k(t) \tag{14}$$

is computed by Algorithm 1.

Next, we (I) sort $\{\boldsymbol{w}_{(k)}\}_k$ by $\text{APPI}_k$ in descending order; (II) output $K$ subsets $\{\mathcal{A}_1, ...\mathcal{A}_K\}$, with the $k$-th subset containing the first $k$ parameter groups.

As a result, we view $\mathcal{A}_k$ as the solution to (11) for any $\epsilon \in \left[\frac{|\mathcal{A}_k|}{|\boldsymbol{w}|}, \frac{|\mathcal{A}_{k+1}|}{|\boldsymbol{w}|}\right)^1$, and as an approximately Pareto optimal solution to (13). Rigorously speaking, by Theorem 2.1, $\mathcal{A}_k$ is not guaranteed to be Pareto optimal, yet it provides empirical guidance for adaptive PEFT design later in Section 5.

## 4 VISUALIZATION OF PARAMETER GROUP INFLUENCE

In this section, we visualize PPI and APPI of (14) in training. We experiment with multiple tasks (image classification, natural language understanding, and generation), model architectures (Vision Transformer or ViT Dosovitskiy et al. (2020), T5 Raffel et al. (2020), RoBERTa Liu et al. (2019), GPT Radford et al. (2019)) and model sizes ($\sim 0.1 - 1B$). We partition models into parameter groups that are fine-tuned by LoRA (Hu et al. (2022); with module names *lora_A* and *lora_B*), BitFit (Zaken et al. (2022); with name *bias*), linear probing (with name *head*), LayerNorm tuning (Zhao et al.; with name *norm*), and embedding layer tuning (with name *embed*).

---

[1]Alternatively, we may use dynamic programming or meet-in-the-middle algorithm algorithm to solve (11) exactly. However, each of these algorithms gives one minimizer of (11), which is not guaranteed to be Pareto optimal in terms of (13) by Theorem 2.1, if the problem has multiple minimizers. Furthermore, to estimate the Pareto frontier, these algorithms need to be applied multiple times, for different $\epsilon$'s, whereas the greed approximation is applied once as it is $\epsilon$-independent.

We generate two types of figures: (I) heatmap of PPI for different parameter groups at different iterations, i.e. $\text{PPI}_k(t)$, where the lighter color indicates a stronger influence; (II) line plot of accumulative PPI, i.e. $\text{APPI}_k(\tau) \approx L(\boldsymbol{w}_0) - L(\boldsymbol{w}_\tau)$. We leave more details in Appendix A.1.

We consistently observe that some parameter groups are highly influential, with $\approx 10^4 \times$ higher PPI than the majority of model parameters. This observation supports the effectiveness of PEFT, only if we actually select the influential parameters.

### 4.1 DISCREPANCY ACROSS MODELS AND TASKS

In Figure 2 and Figure 3, we observe that the influence of parameter groups varies significantly across models and tasks. In summary, none of the methods is performant in all scenarios, which motivates an adaptive PEFT design in Section 5.

**Varying models and tasks**  In Figure 2, it is clear that PPI patterns are highly dependent on model architectures (not sizes) and tasks, and we can leverage such patterns to shed light on the effectiveness of *new PEFT methods*. In fact, it is no surprise that a combination of multiple PEFT methods can give very strong performance. For example, the LoRA library Hu et al. (2022) states that *"training bias vectors in tandem with LoRA might be a cost-efficient way to squeeze out extra task performance"*.

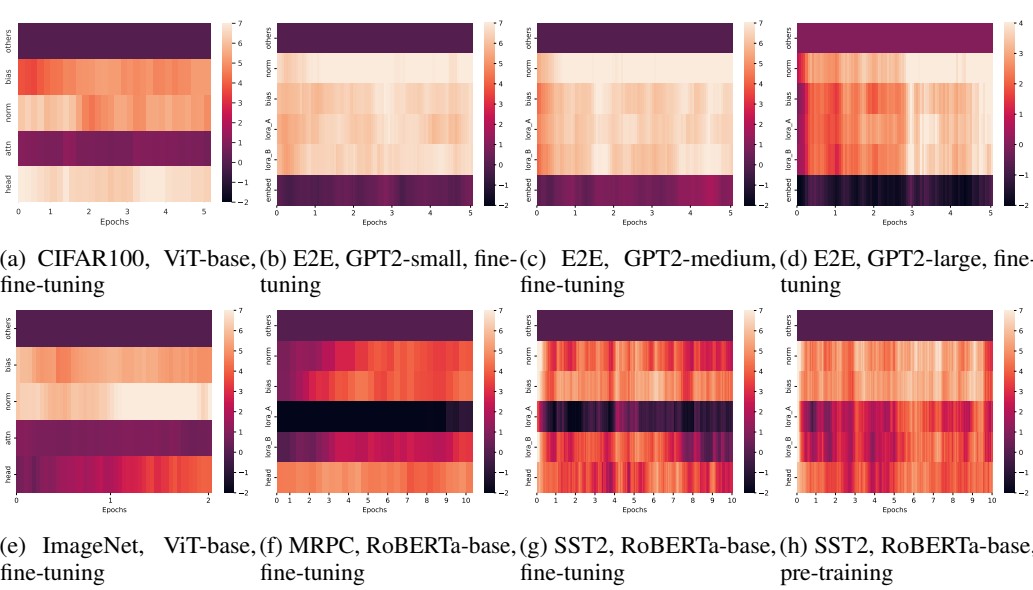

(a) CIFAR100, ViT-base, fine-tuning
(b) E2E, GPT2-small, fine-tuning
(c) E2E, GPT2-medium, fine-tuning
(d) E2E, GPT2-large, fine-tuning

(e) ImageNet, ViT-base, fine-tuning
(f) MRPC, RoBERTa-base, fine-tuning
(g) SST2, RoBERTa-base, fine-tuning
(h) SST2, RoBERTa-base, pre-training

Figure 2: Heatmap of PPI for multiple parameter groups in log-scale.

For example in image classification, on CIFAR100, ViT can be effectively trained with a combination of LayerNorm tuning and linear probing, whereas on ImageNet, only LayerNorm tuning may be sufficient. For example in language modeling, GPT2 on E2E can benefit from a combination of LoRA, BitFit and LayerNorm tuning; RoBERTa-base can be effectively fine-tuned on MRPC and SST2 by linear probing and BitFit, but it may freeze lora_A matrix (like in LoRA-FA Zhang et al. (2023)). In what follows, we demonstrate that PPI can be different by varying only the model or only the task (e.g. dataset or training stage).

**Varying tasks with fixed model**  A closer look at Figure 2(g)(h), both training RoBERTa model on SST2 dataset, reveals that PPI can be different at different training stages: lora_A matrix is less influential in fine-tuning than in pre-training. Additionally, comparing Figure 2(a)(e) or comparing Figure 2(f)(g) reveals that PPI can be different on the same model when the datasets change.

**Varying models with fixed task**  In Figure 3, we compare PPI on T5 and RoBERTa models on the same task. It is clear that different model architectures can have different patterns.

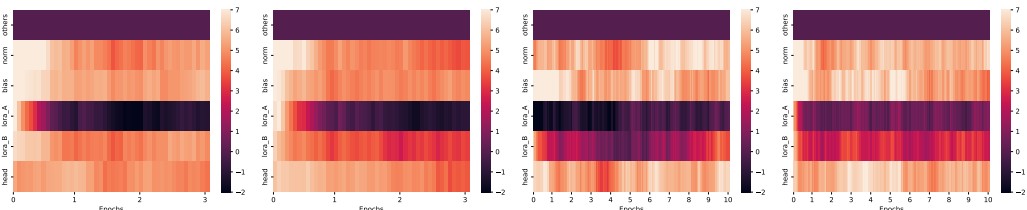

Figure 3: Heatmap of PPI on CoLA in log-scale. Left to right: RoBERTa-base, RoBERTa-large, T5-small, and T5-base.

## 4.2 CONSISTENCY ACROSS TRAINING ITERATIONS

We empirically observe a consistent pattern of PPI and APPI across iterations, shortly after the initialization of models. See Figure 2 and Figure 3 for PPI, and Figure 4 for APPI. Such consistency motivates an efficient strategy to design PEFT: train FMT for some iterations (say 10% of the full run), determine PEFT based on APPI, then launch the full run with PEFT.

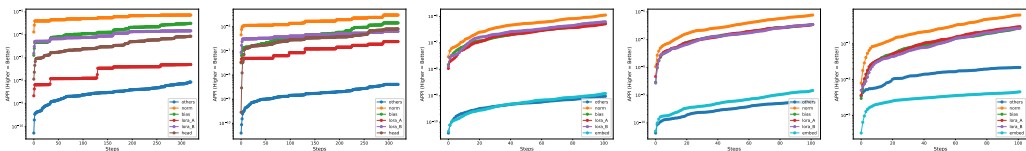

Figure 4: Visualization of APPI. Left two on SST2: RoBERTa-base/large. Right three on E2E: GPT2-small/medium/large.

## 4.3 CONSISTENCY ACROSS MODEL SIZES

Furthermore, we observe a roughly consistent pattern across model sizes for the same architecture and task. We vary RoBERTa and T5 sizes in Figure 3, and GPT2 from small (124M) to large size (0.8B) in Figure 2. We additionally vary ViT from tiny (5M) to large size (0.3B) in Figure 8 in Appendix A. Such observation encourages us to train on small models and directly adopt the optimal PEFT (i.e. the active set of parameter groups) for large models.

## 5 ADAPTIVE PEFT VIA ZERO-SHOT SUBSET TRANSFER

We propose the **AdaPEFT** framework to adaptively select the trainable parameter groups for PEFT.

Firstly, we demonstrate that selecting the active set via APPI is approximately Pareto optimal. In Figure 5 (left column), the theoretical Pareto frontier formed by our selection (6 active sets) closely matches that formed by all $2^6 = 64$ active sets.

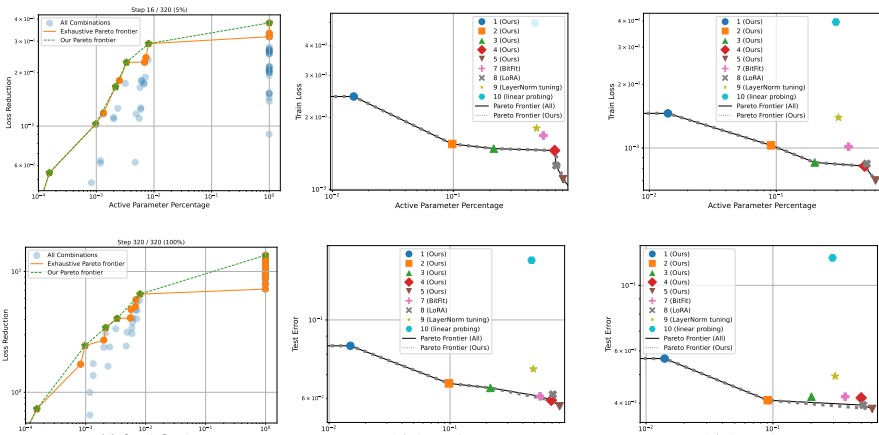

Figure 5: Visualization of Pareto optimality on SST2. Left: theoretical loss reduction of RoBERTa-base via APPI. Middle: actual loss and error of RoBERTa-base. Right: actual loss and error of RoBERTa-large. Each PEFT is indexed in Table 3.

Next, we compare our selection (5 active sets excluding FMT) with existing PEFT in terms of actual train loss and test error. In Figure 5 (middle column), our actual Pareto frontier still closely matches the frontier formed by all 10 methods, whereas some existing PEFT methods are far from the frontier, indicating the potential failure of non-adaptive PEFT.

Having validated the (approximate) Pareto optimality of APPI selection, we now give AdaPEFT in Algorithm 2, transferring the active set of parameter groups from {small models, short training} to {large models, long training} in a zero-shot manner. Note that AdaPEFT can be implemented with much flexibility, e.g. setting $m = M$ or $10\% \to 100\%$.

---

**Algorithm 2** AdaPEFT on model $M$ for $T$ iterations

---

1: Equip a smaller model $m$ with PEFT components (e.g. LoRA and prefix).
2: Train $m$ with Algorithm 1 under FMT, for 10% of $T$ iterations.
3: Sort APPI to select influential parameter groups under $|\mathcal{A}|/|\boldsymbol{w}| \leq \epsilon$ constraint
4: Train $M$ under PEFT with selected $\mathcal{A}$ for $T$ iterations.

---

We experiment with Algorithm 2 on RoBERTa and GPT2. We select the active sets from small models – RoBERTa-base and GPT2-small, using 10% of the training budget, then directly transfer to larger models – RoBERTa-large, GPT2-medium and GPT2-large. We list the active sets, number of parameters and model utility in Table 3 and Table 4 (appendix), which are visualized in Figure 5 and Figure 6.

Our key observation is that (I) the active sets selected by (small model, short horizon) consistently give good Pareto frontier for (large models, long horizon), i.e. AdaPEFT is approximately Pareto optimal and scalable; (II) AdaPEFT effectively evaluates PEFT configurations as strong (if an existing PEFT is close to our Pareto frontier) or weak (if it is far away from the frontier).

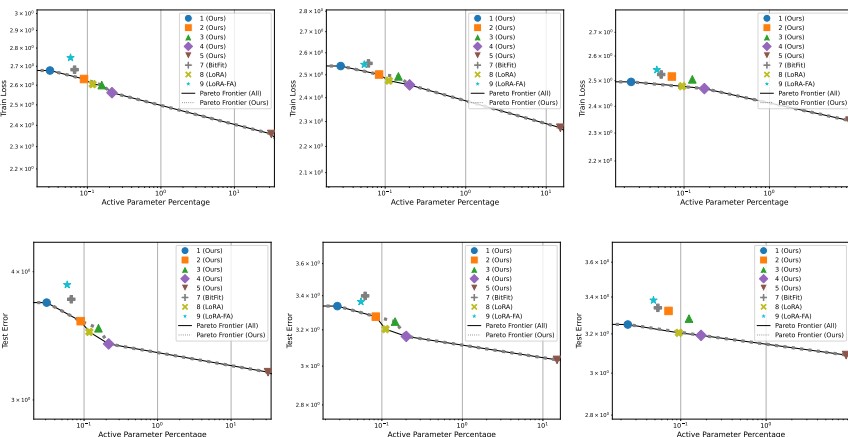

Figure 6: Visualization of Pareto optimality on E2E. Left: actual loss and error of GPT2-small. Middle: actual loss and error of GPT2-medium. Right: actual loss and error of GPT2-large. Each PEFT is indexed in Table 4.

## 6  DISCUSSION

We formulate the selection of active set in PEFT as a multi-task optimization problem, and transform it to 0-1 knapsack problem that we solve under Pareto optimality. In particular, our objective in the knapsack problem is Hessian-informed, which demonstrates that different parameters have different influences on the model performance. Finally, we propose AdaPEFT to leverage such influence pattern and select the active set for large model and long training with minimal budget. We note the success of AdaPEFT depends on the grouping of parameters: a sub-optimal grouping strategy may fail to lead to good performance even with AdaPEFT.

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

# APPENDIX

# A   EXPERIMENT DETAILS

Our experiments are run on A100 GPU, though our approach is independent to the choice of device. Our experiments are three steps: (I) Visualizing the influence under FMT, trained with Algorithm 1 which is updated every 16 iterations (lazy updating); (II) Selecting influential parameter groups to determine PEFT configurations; (III) Training PEFT with GeN AdamW Bu & Xu (2024) (lazy updating frequence 8). For hyper-parameters not mentioned here, we follow Hu et al. (2022).

## A.1   VISUALIZATION METHODOLOGY

Deep learning stochastic optimization is highly non-convex and may be unstable. In addition, the curve fitting approach may occasionally have numerical errors. Therefore, we adopt some outlier removal and smoothing tricks to give reproducible and clear patterns.

For each group $k$ (i.e., a row), our input is a time series of $\text{PPI}_k(t) = \frac{(\mathbf{G}_{(k),t}^{\top}\mathbf{g}_{(k),t})^2}{\mathbf{g}_{(k),t}^{\top}\mathbf{H}_{(k),t}\mathbf{g}_{(k),t}\cdot|\boldsymbol{w}_{(k)}|}$.

For heatmaps and APPI plots, we remove outliers by Interquartile Range (IQR) method [2], smoothen by exponential moving average. For heatmaps, we additionally divide each row by the first row (the *others*, which is majority of parameters). Hence, the first row always stands for one unit of influence. As shown below, our methodology is robust to random seeds in terms of ranking.

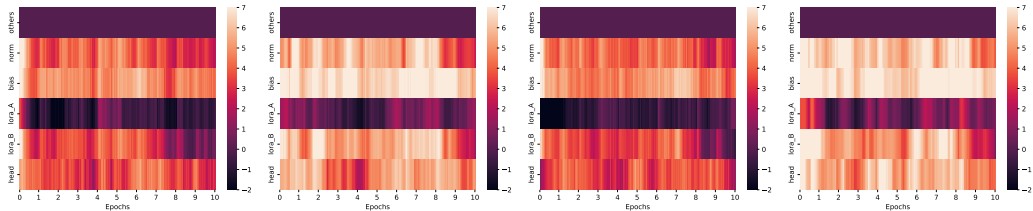

Figure 7: Heatmap of PPI with RoBERTa-base on SST2 dataset from different random seeds.

## A.2   NATURAL LANGUAGE UNDERSTANDING

For NLU tasks, we use batch size 128 and an initial learning rate 2e-5. For CoLA training with RoBERTa, we use a total of 3 epochs. For the rest, we use total epochs of 10. The evaluation metric is test accuracy.

## A.3   GPT2

For GPT2, we experiment on the E2E dataset. For FMT with GeN AdamW, we use initial learning rate 1e-4; for PEFT, it is 1e-3. The sequence length is 128, the total batch size is 256. The total number of epochs for GPT2 (small, medium and large) is 5.

## A.4   ViT CLASSIFICATION

We use ImageNet pre-trained ViT Dosovitskiy et al. (2020), which can be loaded from `timm` library. We resize all images to 224x224 and normalize the pixel values to [-1,1]. We use initial learning rate 1e-4. We apply Algorithm 1 every 16 iterations.

---

[2]The outliers are removed by excluding values outside the range $[Q1 - 3*iqr, Q3 + 3*iqr]$, where the interquartile range $iqr$ is defined as the difference between the 25-th percentile $Q1$ and the 75-th percentile $Q3$ of the data, representing the spread of the central 50%.

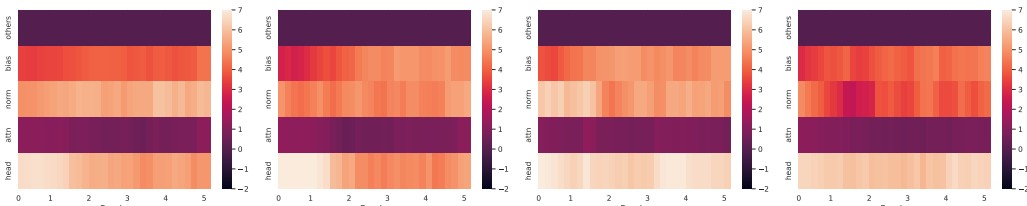

Figure 8: Heatmap of PPI on CIFAR100. Left to right: ViT tiny, small, base and large.

# B EXTENDED EXPERIMENTS

We extend Figure 5 below. Empirically speaking, our AdaPEFT Pareto frontier generated from 6 active sets matches closely the frontier generated from $2^6 = 64$ active sets (all possible combinations) throughout the training.

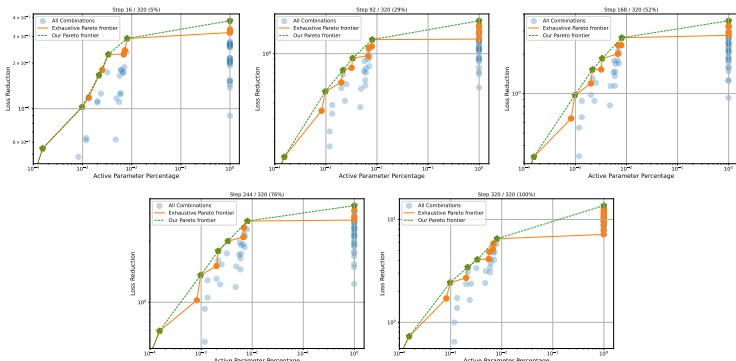

Figure 9: Loss reduction in APPI (see (14)) at different iterations in training. SST2 with RoBERTa-base. 5 epochs, 5270 iterations, logged every 16 iterations by lazy updating ($5270/16 \approx 320$). Dark blue color is due to overlapping.

Furthermore, we reproduce the success of AdaPEFT Pareto frontier on two different architectures, two datasets, and various model sizes.

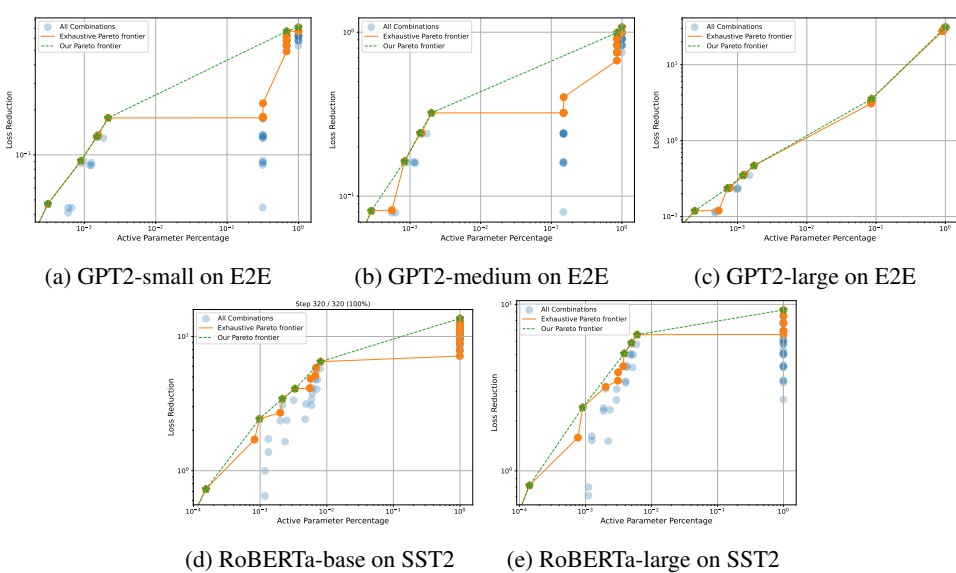

(a) GPT2-small on E2E      (b) GPT2-medium on E2E      (c) GPT2-large on E2E

(d) RoBERTa-base on SST2      (e) RoBERTa-large on SST2

Figure 10: Loss reduction in APPI (see (14)) at the last iteration. Dark blue color is due to overlapping.

## C  TABLES

Table 3: Performance of RoBERTa models on SST2. (Y)es indicates a parameter group is trainable. (N)o indicates a group is frozen.

| model | codename | others | norm | bias | lora_A | lora_B | head | accuracy | % param |
|---|---|---|---|---|---|---|---|---|---|
| RoBERTa-base (ours) | 1 | N | Y | N | N | N | N | $91.55 \pm 0.35$ | 0.015 |
| | 2 | N | Y | Y | N | N | N | $93.39 \pm 0.37$ | 0.098 |
| | 3 | N | Y | Y | N | Y | N | $93.58 \pm 0.41$ | 0.216 |
| | 4 | N | Y | Y | N | Y | Y | $94.07 \pm 0.24$ | 0.689 |
| | 5 | N | Y | Y | Y | Y | Y | $94.30 \pm 0.18$ | 0.807 |
| | FMT | Y | Y | Y | Y | Y | Y | $93.73 \pm 0.66$ | 100 |
| RoBERTa-base (heuristic) | 7(BitFit) | N | N | Y | N | N | Y | $93.92 \pm 0.20$ | 0.556 |
| | 8(LoRA) | N | N | N | Y | Y | Y | $93.85 \pm 0.07$ | 0.709 |
| | 9(LayerNorm) | N | Y | N | N | N | Y | $92.74 \pm 0.57$ | 0.489 |
| | 10(linear probing) | N | N | N | N | N | Y | $85.24 \pm 0.07$ | 0.473 |
| RoBERTa-large (ours) | 1 | N | Y | N | N | N | N | $94.34 \pm 0.78$ | 0.014 |
| | 2 | N | Y | Y | N | N | N | $95.91 \pm 0.13$ | 0.091 |
| | 3 | N | Y | Y | N | Y | N | $95.80 \pm 0.13$ | 0.201 |
| | 4 | N | Y | Y | N | Y | Y | $95.83 \pm 0.29$ | 0.496 |
| | 5 | N | Y | Y | Y | Y | Y | $96.18 \pm 0.18$ | 0.606 |
| | FMT | Y | Y | Y | Y | Y | Y | $95.80 \pm 0.33$ | 100 |
| RoBERTa-large (heuristic) | 7(BitFit) | N | N | Y | N | N | Y | $95.80 \pm 0.29$ | 0.371 |
| | 8(LoRA) | N | N | N | Y | Y | Y | $96.06 \pm 0.26$ | 0.516 |
| | 9(LayerNorm) | N | Y | N | N | N | Y | $95.07 \pm 0.11$ | 0.309 |
| | 10(linear probing) | N | N | N | N | N | Y | $87.61 \pm 0.00$ | 0.295 |

Table 4: Performance of GPT models on E2E. (Y)es indicates a parameter group is trainable. (N)o indicates a group is frozen. We transfer the PEFT identified at $\psi = 10$ to larger models.

| model | codename | others | norm | bias | lora_A | lora_B | embed | perplexity | % param |
|---|---|---|---|---|---|---|---|---|---|
| GPT2-small (ours) | 1 | N | Y | N | N | N | N | $3.73 \pm 0.04$ | 0.031 |
| | 2 | N | Y | N | N | Y | N | $3.58 \pm 0.06$ | 0.09 |
| | 3 | N | Y | Y | N | Y | N | $3.52 \pm 0.03$ | 0.157 |
| | 4 | N | Y | Y | Y | Y | N | $3.40 \pm 0.01$ | 0.216 |
| | 5 | N | Y | Y | Y | Y | Y | $3.19 \pm 0.06$ | 31.827 |
| | FMT | Y | Y | Y | Y | Y | Y | $3.07 \pm 0.00$ | 100 |
| GPT2-small (heuristic) | 7(BitFit) | N | N | Y | N | N | N | $3.76 \pm 0.02$ | 0.067 |
| | 8(LoRA) | N | N | N | Y | Y | N | $3.49 \pm 0.04$ | 0.118 |
| | 9(LoRA-FA) | N | N | N | Y | Y | N | $3.88 \pm 0.01$ | 0.059 |
| GPT2-medium (ours) | 1 | N | Y | N | N | N | N | $3.34 \pm 0.03$ | 0.028 |
| | 2 | N | Y | N | N | Y | N | $3.28 \pm 0.00$ | 0.084 |
| | 3 | N | Y | Y | N | Y | N | $3.25 \pm 0.01$ | 0.146 |
| | 4 | N | Y | Y | Y | Y | N | $3.16 \pm 0.01$ | 0.201 |
| | 5 | N | Y | Y | Y | Y | Y | $3.03 \pm 0.03$ | 14.984 |
| | FMT | Y | Y | Y | Y | Y | Y | $3.03 \pm 0.02$ | 100 |
| GPT2-medium (heuristic) | 7(BitFit) | N | N | Y | N | N | N | $3.40 \pm 0.03$ | 0.062 |
| | 8(LoRA) | N | N | N | Y | Y | N | $3.20 \pm 0.01$ | 0.111 |
| | 9(LoRA-FA) | N | N | N | N | Y | N | $3.36 \pm 0.02$ | 0.055 |
| GPT2-large (ours) | 1 | N | Y | N | N | N | N | $3.25 \pm 0.01$ | 0.024 |
| | 2 | N | Y | N | N | Y | N | $3.32 \pm 0.02$ | 0.072 |
| | 3 | N | Y | Y | N | Y | N | $3.28 \pm 0.02$ | 0.125 |
| | 4 | N | Y | Y | Y | Y | N | $3.19 \pm 0.04$ | 0.173 |
| | 5 | N | Y | Y | Y | Y | Y | $3.09 \pm 0.11$ | 8.645 |
| | FMT | Y | Y | Y | Y | Y | Y | $2.99 \pm 0.04$ | 100 |
| GPT2-large (heuristic) | 7(BitFit) | N | N | Y | N | N | N | $3.34 \pm 0.03$ | 0.054 |
| | 8(LoRA) | N | N | N | Y | Y | N | $3.21 \pm 0.01$ | 0.095 |
| | 9(LoRA-FA) | N | N | N | N | Y | N | $3.39 \pm 0.03$ | 0.048 |

## D  COMPLEXITY ANALYSIS OF ADAPEFT

In this section, we analyze the computational costs and complexity of AdaPEFT in Algorithm 2. For the memory cost, AdaPEFT has the same peak memory as a regular PEFT (e.g. LoRA). Specifically, line 2 (and Algorithm 1) uses a back-propagation-free method to compute the Hessian-informed

selection and the Hessian matrix is never calculated, hence it only adds extra forward passes and time cost, but not the memory cost. We refer the interested readers to Section 4 in Bu & Xu (2024) for details.

For the time cost, there are two parts in Algorithm 2: (I) Algorithm 1 in line 2 on small model and short training, and (II) regular PEFT in line 4 on large model and long training. We argue that, for GPT2-large fine-tuning in Figure 1, part I takes only $\approx 10\%$ and part II takes $100\%$ of the training time of a heuristic PEFT, hence the total training time is $\approx 110\%$. To see this, we analyze the time complexity of Algorithm 1 and compare to the full fine-tuning (FMT), which has two major components: forward pass ($F$) and back-propagation ($B$). A standard FMT costs $F + B \approx 6DN$ Kaplan et al. (2020) where $D$ is the data size (e.g. total number of pixels or tokens) and $N$ is the model size (GPT2-small is 124e6, GPT2-large is 774e6), and roughly $B = 2F = 4DN$. With lazy updating in Section 3.1, running Algorithm 1 costs $(1 + \frac{4K}{\Phi})F + B$, where $\Phi$ is the update frequency for the learning rates. This roughly translates to $(6 + \frac{8K}{\Phi})DN$. To give a quick estimate, when $K = 6, \Phi = 8$, Algorithm 1 is roughly $50\%$ as fast as FMT. Hence, in terms of time complexity,

$$\text{Algorithm 2} = \text{Algorithm 1(small model+short training)} + \text{PEFT(large model+long training)}$$

To be explicit, we analyze the GPT2-large fine-tuning with AdaPEFT. Expetiment settings are the same as Table 4, and we choose the PEFT with codename 4, i.e. mixing LayerNorm tuning, LoRA and BitFit.

FLOP-wise, part I takes $(6 + \frac{8*6}{8}) * (2.7e7 * 10\%) * 124e6 = 7e14$, and part II takes $4*2.7e7*774e6 = 8.4e16$, where we have used the formula that PEFT costs $4DN$ due to not computing most of the parameter gradients. That is, part I takes $< 1\%$ of total time complexity.

We also provide the wall-clock training time. This is different than the theoretical time complexity because of hardware constraints and extra time needed for data loading. Notice that all experiments use the same logical batch size 256 on a single A100 GPU. To maximize GPU memory usage while avoiding out-of-memory during training, we use physical batch size 8 for FMT and 32 for PEFT of fine-tuning GPT2-large and 64 for FMT on GPT2-small.

In summary, AdaPEFT is almost as efficient as regular PEFT.

# E  RELATED WORK

**PEFT methods**  AdaPEFT can work compatibly with many PEFT methods, including those experimented in this work (LoRA and variants, linear probing, BitFit, and LayerNorm), those not experimented (e.g. prompt tuning Lester et al. (2021), prefix tuning Li & Liang (2021), P-tuning Liu et al. (2021), and adapter Houlsby et al. (2019); see Ding et al. (2023); Han et al. (2024); Wang et al. (2024a); Prottasha et al. (2025) for a list of existing PEFT) and those yet-to-come. By taking more PEFT methods into consideration, we allow a larger search space for $\mathcal{A}$ and expect better performance from AdaPEFT, without noticeably increasing the computational cost if we scale the lazy updating accordingly.

Notice that AdaPEFT is a subset selection method that relates closely to subset-based or architecture-wise PEFT methods, and less closely to methods that improve the initialization, optimization and efficiency, such as QLoRA Dettmers et al. (2023), OLoRA Büyükakyüz (2024), LOFTQ Li et al. (2024) , PISSA Meng et al. (2024), LoRA+ Hayou et al. (2024), LoRA-GA Wang et al. (2024b), LoRA-Pro Wang et al. (b). Nevertheless, AdaPEFT is compatible with these methods.

**Multi-task optimization**  Our multi-task formulation in (1) and (13) is solved through $\epsilon$-constraint method. Another standard solution is the linear scalarization or weighted sum method Sener & Koltun (2018); Xin et al. (2022); Kendall et al. (2018): $\min_{\mathcal{A}} L_{\mathcal{A}} + a\frac{|\mathcal{A}|}{|\boldsymbol{w}|}$ for some tunable $a$. Note $\forall a > 0$, without assuming convexity of the objective, this solution is guaranteed to be Pareto optimal. There are two potential challenges: (I) the scalarization problem is usually solved by gradient descent, but our subset selection problem is discrete and non-differentiable; (II) because each $a$ corresponds to one point on Pareto frontier, we need to try a number of $a$, whereas the $\epsilon$-constraint method only needs one sorting. Nevertheless, there may be multi-task optimization methods that can be directly applicable to our problems.

**Hessian-informed loss and learning rate**    We have leveraged Hessian information to formulate our problems. Besides GeN Bu & Xu (2024), a line of work also proposed back-propagation-free ways to compute Hessian-informed learning rate Fu & Wu (2024); Zhu et al. (2021), though the Hessian-informed loss reduction was not presented. It is also possible to use second-order back-propagation (such as Hessian-vector product or Hessian matrix instantiation) to compute the loss reduction we needed. However, this approach will be infeasible unless on very small models.

