# OpenReview forum: "Adaptive parameter-efficient fine-tuning via multi-task optimization on subset selection"
_ICLR.cc/2026/Conference — ICLR 2026 Conference Withdrawn Submission_

### Official Review · Reviewer_BkSU · 2025-10-27

**Soundness:** 2
**Presentation:** 2
**Contribution:** 2
**Rating:** 2
**Confidence:** 4

**Summary:**

The paper proposes AdaPEFT, which frames parameter-efficient fine-tuning as a multi-task problem (utility vs. number of trainable parameters), reduces it via ε-constraint and second-order Taylor expansion to a 0–1 knapsack over parameter groups, and scores each group with a Hessian-informed Per-Parameter Influence (PPI/APPI). A small “probe” run (FMT on a small model for ~10% steps) selects groups; the chosen subset is then transferred to larger models as PEFT. Experiments claim near-Pareto selections and cross-size transferability.

**Strengths:**

1. Clear formalization from bi-objective subset selection to knapsack; neat use of ε-constraint and a closed-form second-order proxy for loss reduction.

2. Practical scoring routine that avoids explicit Hessian-vector backprop; “lazy” updates limit overhead.

3.  Coherent narrative around Pareto frontiers; the visualization of PPI/APPI is informative and may inspire diagnostic tooling.

**Weaknesses:**

1. Approximation stack is fragile/not validated: The chain—ε-constraint ⇒ path-restricted GD ⇒ local 2nd-order fit ⇒ knapsack—rests on strong locality assumptions and positivity of $g^\top H g$; the method discards non-convex neighborhoods by positivity/R² checks, risking biased selection. No stress tests on highly non-convex regimes or adversarial losses.

2. Evaluation scope is dated and light: LLM tests use GPT-2 and small NLU benchmarks; no modern LLMs (LLaMA-2/3, Qwen, Mixtral), no instruction/code/math or long-context settings where PEFT behavior diverges.

3. Baselines are incomplete. Missing comparisons to stronger PEFT families (e.g., DoRA, PiSSA/LoRA+ variants, VeRA, prefix/prompt-tuning at scale, QLoRA-style quantized PEFT). The study largely contrasts against LoRA/BitFit/LayerNorm/linear probing, which is insufficient in 2025/26

4. The wall-clock/Flops plots conflate a small-model FMT “Part I” plus large-model PEFT “Part II” for AdaPEFT, but compare to single-phase LoRA/FMT; it is unclear whether overall tuning budget and data passes are matched.

**Questions:**

1. Why not evaluate layer-wise grouping (per attention/MLP block) to show genuine adaptivity? What happens to runtime and accuracy if K is increased by 10–100?

2. Can APPI be replaced by cheaper proxies (Fisher info, SNIP/GRASP-style saliency, gradient norm) with similar selection? A simple ablation would clarify the value of Hessian fitting.

---

### Official Review · Reviewer_Qzt2 · 2025-10-28

**Soundness:** 2
**Presentation:** 2
**Contribution:** 2
**Rating:** 2
**Confidence:** 3

**Summary:**

The paper introduces AdaPEFT, which searches optimal combinations of multiple parameter-efficient fine-tuning (PEFT) modules per task and model. The authors first discuss a method for formulating the Pareto optimization of the number of parameters (a discrete value) and accuracy (a continuous value) as a 0-1 knapsack problem. The number of parameters is simplified to binary choices—whether to use each of several PEFT methods. The importance of each PEFT method is then estimated, and combinations of PEFT methods are incrementally constructed in the order of importance (using the best PEFT, then using the best and second-best PEFTs, and so on), which are considered to form the Pareto front. To estimate the importance of each PEFT method, the paper proposes a novel approach that approximates the Hessian matrix in the importance factor using only multiple forward calculations, enabling practical computation. Through experiments, the authors claim that their proposed APPI metric can effectively identify Pareto-optimal combinations of PEFT methods.

**Strengths:**

- Mathematically formulating the importance of each PEFT using the Hessian matrix is quite intriguing. While the mathematical transformation might be a common practice in other domains, such as data curation, the mathematical building for PEFT is interesting and will motivate future research.
- The Hessian matrix approximation is an interesting and efficient approach.
- The experimental results are appealing and prove that the proposed approximation works well.

**Weaknesses:**

#### Major Weakness
- Adaptive PEFT itself is not unique, but there are no citations or comparisons with previous methods. For example, S_4 [1], BIPEFT [2], and MambaPEFT [3] utilize evolutionary search methods with several steps to optimize the combination of PEFT methods in addition to the hyperparameters. PrunePEFT [4], AutoPEFT [5], and NOAH [6] utilized supernet-based neural architecture search. The former methods might be time consuming, but the latter methods (PrunePEFT, AutoPEFT, and NOAH) just need the same steps as AdaPEFT.
- The Pareto front discovered by AdaPEFT is not above LoRA, based on Fig. 5 and Fig. 6. It should be important to show the parameter-accuracy trade-off curve of LoRA by changing the rank to prove that AdaPEFT has a better trade-off with other parameter counts.
- The accuracy information of vision models is missing. Also, what pre-trained models are used, especially with the ImageNet finetuning setting?
- Although the formula is interesting, the experiments are very simple, with just deciding whether to use 4 or 5 PEFT methods with pre-defined hyperparameters.

#### Minor Weakness
- The citation style is weird. If you are using LaTeX, it would be better to use \citep{}.
- The statement “some existing PEFT methods are far from the frontier, indicating the potential failure of non-adaptive PEFT” in L434-435 can be incorrect. If PEFT methods far from the frontier are different per task or model, it can be true. However, across all experimental settings, there is a tendency that LoRA is closer to the frontier, while BitFit and LayerNorm tuning are farther from it, suggesting that BitFit and LayerNorm tuning are simply weaker methods.
- Typo
  - L013) only trains -> only train
  - L262) No equation numbering.


[1] Chen, J., Zhang, A., Shi, X., Li, M., Smola, A., & Yang, D. Parameter-Efficient Fine-Tuning Design Spaces. In The Eleventh International Conference on Learning Representations.
[2] Chang, A., Wang, J., Liu, H., Bhatia, P., Xiao, C., Wang, T., & Ma, F. (2024). BIPEFT: Budget-Guided Iterative Search for Parameter Efficient Fine-Tuning of Large Pretrained Language Models. In 2024 Conference on Empirical Methods in Natural Language Processing, EMNLP 2024 (pp. 7429-7440). Association for Computational Linguistics (ACL).
[3] Yoshimura, M., Hayashi, T., & Maeda, Y. MambaPEFT: Exploring Parameter-Efficient Fine-Tuning for Mamba. In The Thirteenth International Conference on Learning Representations.
[4]  Lawton, N. G., Kumar, A., Thattai, G., Galstyan, A., & Ver Steeg, G. (2023, July). Neural Architecture Search for Parameter-Efficient Fine-tuning of Large Pre-trained Language Models. In The 61st Annual Meeting Of The Association For Computational Linguistics.
[5] Zhou, H., Wan, X., Vulić, I., & Korhonen, A. (2024). Autopeft: Automatic configuration search for parameter-efficient fine-tuning. Transactions of the Association for Computational Linguistics, 12, 525-542.
[6] Zhang, Y., Zhou, K., & Liu, Z. (2024). Neural prompt search. IEEE Transactions on Pattern Analysis and Machine Intelligence.

**Questions:**

- Please see major weakness.
- In L262, why is fitting with minus learning rates needed?

---

### Official Review · Reviewer_fTk5 · 2025-10-31

**Soundness:** 3
**Presentation:** 3
**Contribution:** 2
**Rating:** 6
**Confidence:** 3

**Summary:**

This paper introducesa technique to adaptively select the most effective subsets of parameters for fine-tuning. The key idea is that they formulate the subset selection of parameters as a multi-task optimization problem consisting of a discrete objective ( over the subsets) and a continuous objective (minimization of loss). The goal is to maximize model performance while minimizing the number of trainable parameters. This problem is then transformed into a 0-1 knapsack problem, where the "value" of each parameter group is its potential for loss reduction, calculated using Hessian (second-order) information. The key finding is that these "influential" parameter groups can be identified early in training on small models and effectively transferred to optimize the fine-tuning of large models.

**Strengths:**

This is an interesting paper and the formulation makes sense. To me, the strengths are
  1. the formulation of the multi-task optimization problem (ie maximizing performance and minimizing trainable model parameters) as a classical 0-1 knapsack problem is nice and an approach that is novel and appealing.
  2. the authors use a Hessian-informed analysis to measure the value of a different parameter groups. This is a nice principled and data-motivated way to determine the most important parameters for fine-tuning.
  3. Their framework is scalable to large models and not expensive in terms of compute budget

**Weaknesses:**

- If I understand correctly, their method AdaPEFT relies on groupings of parameters. So that, a suboptimal grouping strategy may fail to lead to good performance. However, this technique can't determine influential parameters within a group or determine new, more advantageous smaller groupings.
 - This method relies on the Hessian to determine loss-reduction. The Hessian is second order and the algorithm requires multiple forward passes (if I understand, $4k+1$ passes where $k$ is the number of subset parameter groups). This is quite a lot of computational overhead compared to other off-the-shelf PEFT methods.

**Questions:**

I have a few questions related to the weaknesses described above

 - Is it correct that this technique performs some kind of optimization on the group level but not within group level. How does this hinder or affect results? How sensitive are the results to the choice of parameters subsets from the beginning.
 - What is the comparison of computation required for AdaPEFT vs the other PEFT methods described in the paper; eg in terms of FLOPs? As mentioned above, the Hessian approximations require multiple forward passes. I understand the performance improvements but how much do these come at a cost of computation?

---

### Note · Authors · 2025-11-24

I have read and agree with the venue's withdrawal policy on behalf of myself and my co-authors.